# Quantifying unmet need in General Practice: a retrospective cohort study of administrative data

Alex McConnachie [ID],[1] David A Ellis [ID],[2] Philip Wilson [ID],[3] Ross McQueenie,[4] Andrea E Williamson [ID] [5]

[1]Robertson Centre for Biostatistics, School of Health and Wellbeing, University of Glasgow, Glasgow, UK
[2]School of Management, University of Bath, Bath, UK
[3]Institute of Health and Wellbeing, University of Aberdeen, Aberdeen, UK
[4]Place and Wellbeing Directorate, Public Health Scotland, Edinburgh, UK
[5]General Practice and Primary Care, School of Health and Wellbeing, University of Glasgow, Glasgow, UK

**Correspondence to**
Professor Alex McConnachie;
Alex.McConnachie@glasgow.ac.uk

## ABSTRACT

**Objectives** To assess whether patients attending general practices (GPs) in socioeconomically (SE) deprived areas receive the same amount of care, compared with similar patients (based on age, sex and level of morbidity) attending GPs in less deprived areas. If not, to quantify the additional resource that would be required by GPs in deprived areas to achieve parity.

**Design** Retrospective cohort study.

**Setting** 150 GPs in Scotland, UK, divided into two groups: 80 practices in Scottish Index of Multiple Deprivation (SIMD) deciles 1–5 (more SE deprived); 70 practices in SIMD deciles 6–10 (less SE deprived).

**Patients** 437 590 patients registered with a more SE deprived GP, and 333 994 patients registered with a less SE deprived GP, for the whole study period (2013–2016), who made at least one appointment.

**Outcomes** The number of contacts and total contact time between patients and clinical staff.

**Results** Patients in more SE deprived areas had slightly more discrete contacts over 3 years (11.8 vs 11.4), but each patient had marginally less contact time (146.1 vs 149.5 min). Stratified by sex and age, differences were also small. Stratified by the number of long-term conditions (LTCs), practices in more SE deprived areas delivered significantly less contact time than practices in less SE deprived areas. Over 3 years, 8 fewer minutes for patients with no LTCs, and 24, 27, 38 and 28 fewer minutes for patients with 1, 2, 3–4 or 5+LTCs, respectively.

**Conclusion** If GPs in more SE deprived areas were to give an equal amount of direct contact time to patients with the same level of need served by GPs in less SE deprived areas, this would require a 14% increase in patient contact time. This represents a significant unmet need, supporting the case for redistribution of resources to tackle the inverse care law.

## STRENGTHS AND LIMITATIONS OF THIS STUDY

⇒ Large dataset, based on routine administrative records, demonstrating the feasibility of measuring contact time with practice staff.
⇒ Large sample of practices covering urban and rural, affluent and deprived areas.
⇒ The accuracy of time spent with patients based on the opening and closing of medical records has not been verified.
⇒ Healthcare need is estimated by a simple count of the number of long-term conditions, which does not reflect variation in healthcare needs between conditions, nor variation in needs between patients within the same condition.
⇒ These data predate the rapid and far-reaching changes to general practice access in the UK triggered by the COVID-19 pandemic.

## INTRODUCTION

Unmet need is a multidimensional, complex construct;[1] '*when an individual does not receive an available and effective treatment that could have improved her health…some unmet need is acceptable since resources are scarce. What is of concern here is whether unmet need is inequitable, or systematically related to socioeconomic or other personal characteristics*'.[2]

Patients living with multimorbidity (two or more coexisting long-term health conditions (LTCs)) have complex needs, and successfully meeting those within our healthcare systems is one of the most pressing challenges of our time.[3–5] We already have strong evidence that people living in more socioeconomically (SE) deprived areas in the UK have a burden of multimorbidity at an earlier age.[6] Furthermore, patients with multimorbidity are at greater risk of all-cause mortality[7 8] and hospitalisation,[8] regardless of whether the number of LTCs are measured in a research setting or via primary care records,[9] suggesting that the numbers of LTCs recorded in general practice (GP) electronic health records broadly reflect underlying healthcare need.

The analyses presented in this paper stem from research exploring the determinants and outcomes of missed appointments in GP in Scotland. During this work, we determined the total number and duration of contacts between clinical practice staff and individual patients. We postulated that patients with more LTCs have greater healthcare need, and therefore, require more contact time

with clinical practice staff. In an equitable health service, equivalent patients (according to their age, sex and number of LTCs) should, on average, receive the same amount of contact with clinical practice staff, regardless of whether they live in an affluent or SE deprived area. Therefore, we hypothesised that patients in SE deprived communities may receive less clinical GP staff contact time with an equivalent level of morbidity. If this is the case, how much more contact time would be required in SE deprived practices to achieve parity with affluent practices?

## METHODS

A trusted third party (TTP) for the National Health Service (NHS) in Scotland (Albasoft), recruited GPs on our behalf. A TTP is an organisation independent from the NHS with the skills, infrastructure and permissions to extract and link data safely without having a direct interest in the use to which the data will be put. The TTP is required to ensure confidentiality for patients and professionals in the research process.[10] The TTP recruited practices to ensure a spread of practices in urban and rural locations, serving both affluent and SE deprived (Deep End) populations. All patients who were alive and registered at the same practice for the 3-year period (5 September 2013 to 5 September 2016) and who made at least one appointment with the practice were included in the data set. All patients retained also had data on age, sex, number of recorded LTCs, and the number and total duration of all in person contacts with clinical practice staff such as GPs and practice nurses. Recording of patient ethnicity and language interpreter requirements was very low so this could not be included in our analysis.[11]

How we defined and measured LTCs are documented in previous publications.[11–13]

We present the number and percentage of patients in practices serving affluent and deprived populations, according to sex, age (0–39, 40–59, 60–79 and 80+ years) and the number of LTCs (none, 1, 2, 3–4 or 5+). Practices were categorised as affluent or deprived based on the average Scottish Index of Multiple Deprivation (SIMD) score of all patients registered at the practice. SIMD is routinely used in Scotland as part of the current GP contract funding allocation formula. When split by decile groups, 1 is the most SE deprived.[14]

We calculated the total number and duration of contacts in minutes for each patient. These were extracted from appointments recorded via practice IT systems. The duration of contacts was calculated from the 'in' and 'out' time. Detailed proof-of-concept work in our pilot study of 67 705 patient records enabled us to set rules about included appointments to increase accuracy. Appointments that were 'open' for less than 2 min, 'open' for more than 24 hours, labelled as an administrative slot, or were not face-to-face appointments were excluded.[12] There is no unified categorisation of GP appointment types in the UK and this was reflected in the heterogeneity of appointment categories in our data set. Therefore, all appointments included were for face-to-face appointments with a healthcare professional in the practice (rather than being specific, eg, about GP or practice nurse contacts).[11]

We present the mean and SD of the number and duration of contacts for patients in practices in affluent and deprived areas, according to sex, age and the number of LTCs.

Taking the mean contact time provided by practices in affluent areas and applying this to the number of patients served by practices in SE deprived areas, we calculated the change in total contact time that would be required in practices in SE deprived areas in order to give the same average contact time. We then repeated this analysis, stratified by sex, age and/or number of LTCs.

All analyses were done using R statistical software.[15]

### Patient and public involvement

The Royal College of General Practitioners Scotland Patient Partnership in Practice (P³) Committee in 2016 provided advice about the relevance and importance of this research. This was to inform the study team's application for data linkage of routine data sets. The P³ committee (a lay patient group) advised that they agreed it was 'interesting, much needed and beneficial' and they have been updated on progress as the project has proceeded.

## RESULTS

Data from 150 Scottish practices were included. These were split into two groups: SIMD decile 1–5 (most SE deprived, 80 practices) and SIMD decile 6–10 (least deprived, 70 practices). From a cohort of 824 374 patients, after exclusion of patients who moved practice, or who had missing data for age, sex or number of LTCs, data from 771 584 patients' records were analysed. The study population was 437 590 patients from the 80 practices in SE deprived areas, and 333 994 patients from the 70 practices in affluent areas. Table 1 sets out their characteristics. Differences in terms of the sex and age distributions are small. Forty-seven per cent of patients from SE deprived practices in this sample were male, compared with 46% of patients in practices in more affluent areas. Patients registered with practices in SE deprived areas were slightly younger, with 48% vs 47% being under 40, and 4.6% vs 5.2% being over 80 years of age. The difference in terms of the distribution of the number of LTCs that patients had was far more marked, with 45% of patients in practices serving affluent populations having no LTCs, compared with 37% of patients in practices serving SE deprived populations.

Table 2 shows the mean number and duration of contacts between patients and clinical practice staff, by age, sex and number of LTCs, and the estimated differences between practices serving a population living in relatively SE deprived areas, and practices serving more

**Table 1** Study population

|  |  | Practices in deprived areas | Practices in affluent areas |
|---|---|---|---|
| No of practices |  | 80 | 70 |
| No of patients |  | 437 590 | 333 994 |
| Sex |  |  |  |
| Male | N (%) | 205 914 (47.1) | 155 289 (46.5) |
| Female | N (%) | 231 676 (52.9) | 178 705 (53.5) |
| Age group |  |  |  |
| 0–39 years | N (%) | 211 917 (48.4) | 157 731 (47.2) |
| 40–59 years | N (%) | 123 385 (28.2) | 93 368 (28.0) |
| 60–79 years | N (%) | 82 315 (18.8) | 65 559 (19.6) |
| 80+ years | N (%) | 19 973 (4.6) | 17 336 (5.2) |
| No of long-term conditions |  |  |  |
| 0 | N (%) | 163 616 (37.4) | 151 063 (45.2) |
| 1 | N (%) | 98 470 (22.5) | 74 497 (22.3) |
| 2 | N (%) | 63 847 (14.6) | 43 597 (13.1) |
| 3–4 | N (%) | 72 814 (16.6) | 44 006 (13.2) |
| 5+ | N (%) | 38 843 (8.9) | 20 831 (6.2) |

Numbers and percentages of patients in groups defined by sex, age and number of long-term conditions, for practices with SE deprived and affluent populations.
SE, socioeconomically.

affluent areas. Confidence intervals are derived from simple two-sample t-tests. Figure 1 displays mean contact time by age, sex and number of LTCs, for practices with SE deprived and affluent populations.

Overall, patients in SE deprived areas had more discrete contacts with clinical practice staff over 3 years (11.8 vs 11.4), but each patient had less contact time (146.1 vs 149.5 min). Men in SE deprived areas had the same number of contacts, but spent 7.4 min less time with clinical staff, compared to men in affluent areas. Women in practices in SE deprived areas had slightly more contacts than women in affluent area practices (0.8 more contacts over 3 years); but spent a similar amount of time with clinical practice staff.

Patterns by age are more complex. For the youngest age group (under 40 years), and for those aged 60–70 years, patients in SE deprived area practices had marginally more contacts, but less time with clinical practice staff. In the 40–59 age band, however, patients in SE deprived areas had slightly more contacts, and spent slightly more time with clinical practice staff. In the oldest patients (over 80 years), patients registered to practices serving more SE deprived populations had fewer contacts and spent less time with the practice.

However, when looking at the data in relation to the number of LTCs that were recorded, there is a consistent and striking pattern. Patients in practices in SE deprived areas clearly have fewer contacts, and spend less time with clinical practice staff, compared with patients in more affluent practices, at every level of morbidity.

Table 3 shows how much contact time would be required between patients and staff in practices serving more SE deprived populations if the average contact time were the same as in practices with more affluent populations. Overall, practices in affluent areas spent about 50 million minutes with 334 000 patients, at an average of roughly 150 min per patient over the 3-year study period. This was slightly more than the average contact time in practices in

**Table 2** Mean number and duration of appointments (over 3 years), by age, sex and number of long-term conditions, for practices with SE deprived and affluent populations

|  | Total no of appointments | | | Total duration of appointments (min) | | |
|---|---|---|---|---|---|---|
|  | Practices in deprived areas | Practices in affluent areas |  | Practices in deprived areas | Practices in affluent areas |  |
|  | Mean (SD) | Mean (SD) | Difference (95% CI), p value | Mean (SD) | Mean (SD) | Difference (95% CI), p value |
| All | 11.8 (14.6) | 11.4 (14.2) | 0.4 (0.3 to 0.5), p<0.0001 | 146.1 (228.4) | 149.5 (223.4) | −3.3 (−4.4 to −2.3), p<0.0001 |
| Sex |  |  |  |  |  |  |
| Male | 9.9 (13.2) | 9.9 (13.3) | 0.0 (−0.1 to 0.1), p=0.4361 | 121.1 (207.0) | 128.5 (209.6) | −7.4 (−8.8 to −6.0), p<0.0001 |
| Female | 13.5 (15.6) | 12.7 (14.8) | 0.8 (0.7 to 0.9), p<0.0001 | 168.4 (243.7) | 167.7 (233.2) | 0.7 (−0.8 to 2.2), p=0.36 |
| Age |  |  |  |  |  |  |
| 0–39 years | 8.2 (9.6) | 7.6 (8.9) | 0.5 (0.5 to 0.6), p<0.0001 | 95.2 (135.1) | 97.6 (143.4) | −2.4 (−3.3 to −1.5), p<0.0001 |
| 40–59 years | 12.7 (14.7) | 11.3 (13.3) | 1.3 (1.2 to 1.5), p<0.0001 | 157.6 (222.6) | 153.9 (223.4) | 3.8 (1.9 to 5.7), p=0.0001 |
| 60–79 years | 17.6 (18.6) | 17.3 (18.1) | 0.3 (0.1 to 0.5), p=0.0021 | 223.9 (299.5) | 225.9 (284.8) | −2.0 (−5.0 to 0.9), p=0.18 |
| 80+ years | 21.8 (24.1) | 24.3 (23.5) | −2.5 (−3.0 to −2.0), p<0.0001 | 294.5 (445.1) | 308.1 (356.6) | −13.6 (−21.7 to −5.4), p=0.0011 |
| No of long-term conditions |  |  |  |  |  |  |
| 0 | 5.9 (6.7) | 6.0 (6.5) | −0.1 (−0.2 to −0.1), p<0.0001 | 66.9 (95.6) | 74.7 (101.5) | −7.9 (−8.6 to −7.2), p<0.0001 |
| 1 | 9.2 (9.4) | 10.1 (9.9) | −0.8 (−0.9 to −0.7), p<0.0001 | 107.0 (134.7) | 130.8 (157.7) | −23.8 (−25.3 to −22.4), p<0.0001 |
| 2 | 13.4 (13.0) | 14.1 (13.1) | −0.7 (−0.8 to −0.5), p<0.0001 | 161.1 (195.9) | 187.9 (223.1) | −26.8 (−29.4 to −24.2), p<0.0001 |
| 3–4 | 18.8 (17.5) | 20.5 (18.4) | −1.7 (−1.9 to −1.5), p<0.0001 | 236.8 (267.6) | 275.0 (305.2) | −38.2 (−41.7 to −34.8), p<0.0001 |
| 5+ | 27.7 (25.2) | 30.7 (26.9) | −3.0 (−3.5 to −2.6), p<0.0001 | 384.6 (450.3) | 412.3 (434.1) | −27.7 (−35.1 to −20.3), p<0.0001 |

SE, socioeconomically.

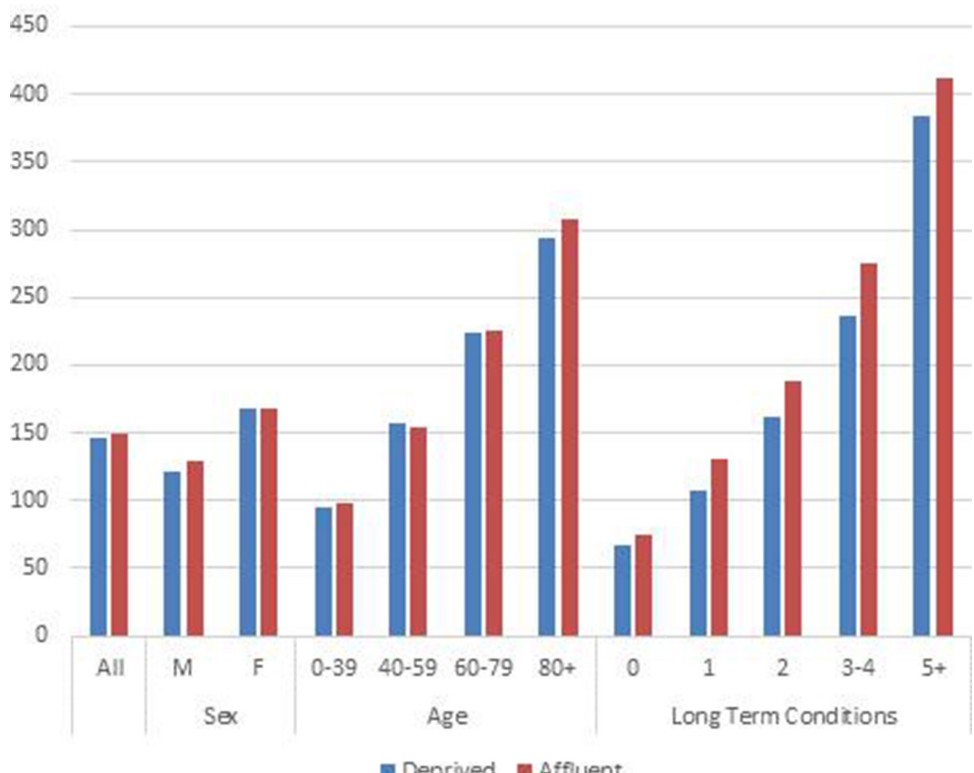

**Figure 1** Mean duration of appointments (over 3 years), by age, sex and number of long-term conditions, for practices with SE deprived and affluent populations. SE, socioeconomically.

SE deprived areas, so that if these practices were to have the same average contact time, the total amount of time spent in contact with patients would need to increase by 2.3%.

The same calculations can be stratified by sex. The increase in contact time required in practices in SE deprived areas appears to be limited to male patients, but when the two sexes are combined, the adjustment required is similar to the overall calculation, at 2.1%. Broken down by age, different age bands are seen to require different adjustments, some positive and some negative, so that overall, the increase required in practices in SE deprived areas appears minimal, at 0.8%.

A different picture appears when stratifying the study population by the number of LTCs. At every level of morbidity, patients in practices in SE deprived areas have less contact time with their practices, and practices would require a significant increase in resources to deliver the same amount of care as practices in more SE affluent areas. The smallest increase required is 7.2%, for patients with the most complex needs, with five or more LTCs. When combined, these adjustments add up to an increase in contact time required in SE deprived areas of 14.4%.

Each of these calculations is stratified by only one condition, but the same calculations can use combinations of stratification factors, and these are shown in table 3. Stratifying by age and sex alone, reveals that very little adjustment is required to achieve parity in terms of mean

contact time. However, as soon as the number of LTCs are factored into the calculations, it is clear that a large increase in resources would be required for practices in SE deprived areas to offer the same amount of contact time to patients with the same level of need. Stratifying by age, sex and number of LTCs, the estimated adjustment required is an increase of 13.5%.

## DISCUSSION
### Summary
Using patient records from a sample of 150 Scottish GPs, we demonstrate that substantially less care is received by patients in practices in SE deprived areas compared with patients in affluent areas, but only when viewed in relation to morbidity, as measured by the number of LTCs. If GPs in highly SE deprived areas were to give an equivalent amount of time (direct contact) to patients with the same level of need (the number of long-term conditions), they would require additional resources sufficient to support an increase of approximately 14% in patient contact time.

### Strengths and limitations
This study uses routine administrative data from a large population dataset which limits the potential bias associated with self-reported utilisation data[2] when seeking to measure unmet need. It demonstrates that data are retrievable from practice data systems to measure contact

**Table 3** Additional resource requirements (in terms of total appointment time) for general practices in deprived areas to offer equal appointment times to general practices in affluent areas

| Stratification | Practices in affluent areas (N=70) | | Practices in deprived areas (N=80) | | | |
| | Observed appointment times | | Observed appointment times | | Assuming equitable appointment times | |
| | Total minutes | Mean | Total minutes | Mean | Total minutes | % Change required |
|---|---|---|---|---|---|---|
| None | 49 919 411 | 149.5 | 63 942 072 | 146.1 | 65 403 076 | +2.3 |
| Sex—male | 19 951 031 | 128.5 | 24 930 987 | 121.1 | 26 455 168 | +6.1 |
| Sex—female | 29 968 380 | 167.7 | 39 011 085 | 168.4 | 38 851 484 | 0.4 |
| Sex—overall | | | | | 65 306 652 | +2.1 |
| Age—0–39 | 15 401 793 | 97.6 | 20 180 456 | 95.2 | 20 692 836 | +2.5 |
| Age—40–59 | 14 365 312 | 153.9 | 19 450 870 | 157.6 | 18 983 635 | 2.4 |
| Age—60–79 | 14 811 401 | 225.9 | 18 428 732 | 223.9 | 18 596 996 | +0.9 |
| Age—80+ | 5 340 905 | 308.1 | 5 882 014 | 294.5 | 6 153 317 | +4.6 |
| Age—Overall | | | | | 64 426 784 | +0.8 |
| LTC count—0 | 11 291 456 | 74.7 | 10 942 172 | 66.9 | 12 229 751 | +11.8 |
| LTC count—1 | 9 744 067 | 130.8 | 10 531 390 | 107.0 | 12 879 690 | +22.3 |
| LTC count—2 | 8 192 478 | 187.9 | 10 285 517 | 161.1 | 11 997 732 | +16.6 |
| LTC count—3–4 | 12 103 247 | 275.0 | 17 244 417 | 236.8 | 20 026 492 | +16.1 |
| LTC count—5+ | 8 588 163 | 412.3 | 14 938 576 | 384.6 | 16 014 114 | +7.2 |
| LTC count —overall | | | | | 73 147 780 | +14.4 |
| Age, sex—overall | | | | | 64 311 097 | +0.6 |
| Sex, LTC count—overall | | | | | 72 988 341 | +14.1 |
| Age, LTC count—overall | | | | | 72 834 600 | +13.9 |
| Age, sex, LTC count—overall | | | | | 72 563 329 | +13.5 |

LTC, long-term condition.

time with practice staff and allow for an estimation of resources required. The findings begin to address a universal problem about unmet need and resource allocation in primary care. However, our analysis excludes patients who made no appointments with their practice over the 3-year period. This may slightly bias our estimates of the average time spent with practice staff. However, such patients are likely to have very low healthcare need and will have little impact on the resource requirements of their practices.

The accuracy of estimated time spent with patients has also not been ascertained in research, but we have reduced any inaccuracies by using record opening times to best reflect clinical contacts.[12] Outside the Quality and Outcomes Framework,[16] there is also some variation in how LTCs are coded. However, we followed established methods to mitigate for potential errors.[6] We acknowledge that each LTC included does not have equivalence in terms of impact on patients and that even within each condition there will be variation in the healthcare needs of individuals.[17]

The measure of socioeconomic deprivation used is area based, rather than one measured at an individual level to maximise accuracy; however, this was the best available. Likewise we were unable—due to the anonymised nature

of practice recruitment—to recruit a sample that was a perfect representation of the population; however, our TTP recruited practices based on our specific request to ensure an urban/rural and affluent/SE deprived area spread.

The lack of accurate information about ethnicity is a limitation. In our previous work,[11] we found that only 2.7% of appointments could be linked to ethnicity data in the medical records.

Given our aim of highlighting a potential source of unmet need, we used a simple division of practices into those serving relatively more SE deprived and affluent populations. Whether these findings hold true on a national scale, across the full spectrum of SE deprivation, remains to be seen.

Also, given the known association between deprivation and multimorbidity, our analysis focused on SE deprivation. Provision of medical care by GPs does have some distinct features depending on practice locality. For example, in remote rural settings practices provide more prehospital emergency care which means more time may be spent with patients in those circumstances and this may bias consultation time estimates in these areas. Future work will need to consider the impact of rurality on the associations we have observed.

Finally, these data predate the rapid and far-reaching changes to GP access in the UK triggered by the COVID-19 pandemic. It will be some time yet before the longer-term consequences and eventual access arrangements settle into a new normal and their consequences are understood.

### Comparison with existing literature

Research from Scotland over 20 years ago, and more recently, observed that consultations in more SE deprived areas tended to be shorter[18 19] and these findings have been replicated in England.[20] We know that provision of GP resource is flatly distributed in Scotland[21] and there are indications that the revised Scottish GP contract funding formula from 2018 has not had a positive impact in more SE deprived areas.[22] An in-depth analysis conducted with English routine administrative data showed that practices in high SE deprivation areas have fewer GPs and more practice nurses than affluent area-based counterparts. This means GPs in those settings are caring for 10% more patients overall.[23] What this paper demonstrates, for the first time, is how clinically focused evidence that relies on high quality data from a large percentage of the Scottish population can inform and improve models of care. Put simply, it appears that on average, a patient registered at a GP in Scotland serving an affluent population will receive more care from their practice than a patient of the same age, sex and level of morbidity, registered with a practice serving a more SE deprived population.

### Implications for research and/or practice

This study adds to growing evidence that provision of GP care in the UK is not equitably distributed across practices and needs to change.[24] Patients with LTCs living in the least affluent areas are losing out. This is unacceptable unmet need because it is systematically related to socioeconomic factors.

Based on a relatively simple analysis using a sample of practices in Scotland, we estimate that a 14% increase in clinical staff contact time is required in SE deprived areas to equalise provision. A direct impact of this analysis is that governments, health service policy- makers and planners should consider using data on contact time to measure the amount of care delivered to patients, as well as data on the number of recorded LTCs to estimate the level of need of GP practice populations. This is now achievable through routine administrative data capture and could be factored into resource allocation algorithms.

This paper adds to the large body of work now accumulated in the UK about the role GP care has in perpetuating health inequalities and the inverse care law.[25] If our findings are replicated at a national level, using more complex analysis methods as used in deriving resource allocation formulas, then GP could start to become part of the solution to socioeconomic health inequalities.

### CONCLUSION

GP provision in SE deprived areas needs to be better supported to deliver uniform care to patients with the same level of need across the UK. The analysis from this paper suggests that measuring LTCs and funding accordingly would make a significant contribution to achieving this aim.

**Correction notice** This article has been corrected since it was published. The licence has been updated to CC BY on 25th March 2024.

**Acknowledgements** Thank you to all the GP practices who participated in the Serial Missed Appointments study and for strategic support from Ellen Lynch (Health and Social Care Analytical Services, Scottish Government). The general practice data expertise of Dave Kelly (Albasoft) was invaluable. Thanks also to the eDRIS team who facilitated the safe use of our data in the Safehaven, especially Dionysis Vragkos.

**Contributors** AEW: designed and managed the study, wrote first draft of manuscript, reviewed final version of manuscript. DAE and PW: designed and managed the study, reviewed final version of manuscript. RM: processed data for analysis, reviewed final version of manuscript. AM: conceived and carried out statistical analysis, wrote statistical methods and results, reviewed final version of manuscript and is responsible for the overall content as guarantor.

**Funding** This work was supported by a grant from Chief Scientist Office, Scottish Government (reference CZH/4/1118) with Safe Haven and data linkage costs supported in lieu by the DSLS at Scottish Government.

**Disclaimer** The funding sources for this analysis had no influence over study design, data collection, data analysis, data interpretation, the writing of the report, or the decision to submit for publication.

**Competing interests** None declared.

**Patient and public involvement** Patients and/or the public were involved in the design, or conduct, or reporting, or dissemination plans of this research. Refer to the Methods section for further details.

**Patient consent for publication** Not applicable.

**Ethics approval** Letters of comfort were issued by the West of Scotland NHS Ethics Committee and the University of Glasgow College of Medical, Veterinary & Life Sciences Ethics Committee confirming that the full study did not need NHS ethics approval as this used routine data that had been extracted in an anonymised format. Public Benefit and Privacy Panel approval for use of the data was granted by NHS Information Services Scotland in December 2016.

**Provenance and peer review** Not commissioned; externally peer reviewed.

**Data availability statement** Data may be obtained from a third party and are not publicly available. These data were available from NHS Scotland and access permissions were contingent on them not being publicly available. New data access and TTP requests can be made to www.escro.co.uk for general practice data and https://www.isdscotland.org/products-and-services/edris/ to host the data securely and request data linkage. The authors had no special access permissions that others could not have. Analysis code is available from the authors upon reasonable request. The code will replicate the study findings, but the analysis can be replicated using other statistical software.

**ORCID iDs**
Alex McConnachie http://orcid.org/0000-0002-7262-7000
David A Ellis http://orcid.org/0000-0001-6172-3323

Philip Wilson http://orcid.org/0000-0002-4123-8248
Andrea E Williamson http://orcid.org/0000-0002-8981-9068

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
