## [Reviewer comments · BMJ Open]

ARTICLE DETAILS

TITLE (PROVISIONAL)	Quantifying unmet need in General Practice: a retrospective cohort study of administrative data
AUTHORS	McConnachie, Alex; Ellis, David; Wilson, Philip; McQueenie, Ross; Williamson, Andrea

VERSION 1 – REVIEW

REVIEWER	Graham Kirkwood Newcastle University, Institute of Health and Society
REVIEW RETURNED	24-Oct-2022

GENERAL COMMENTS	lines 81 to 83, I don't understand how the conclusion reached at the end of the sentence is warranted by what goes before Table 1. Why not do some statistical testing here to establish whether the groups are different? Table 2. instead of or in addition to saying "over 3 years" could the authors give the actual dates Line 204. It might be worth a sentence saying this is only measuring the unmet need of those who actually turn up at GPs, there will be other levels of unmet need in the community unable to be measured here
---

REVIEWER	Jesse Whitehead University of Waikato
REVIEW RETURNED	29-Nov-2022

GENERAL COMMENTS	Thank you for the opportunity to review this important and interesting paper. Overall, the paper is of a high standard and I enjoyed reading and reviewing it. There are however some key points that need to be addressed. Level of care: First, there are several conceptual issues that need to be clarified or appropriately addressed. One issue is that the outcomes of interest (number of contacts and total contact time) are very often conflated with "level of care" as outlined in the study objectives, or "unmet need" as described in the first sentence of the introduction. Although there is likely to be a correlation between contacts and contact time with unmet need / level and quality of care, it is important to clearly state that this paper is not directly measuring the 'level', 'quality', or 'effectiveness' of treatment provided in different practices. The big assumption of this whole
---

	paper is that “time-spent” and “number of contacts” = better quality treatment, but I’m not sure if that has been clearly articulated with sufficient supporting evidence. Socioeconomic position: Another key issue is that socio-economic deprivation is not clearly conceptualised, and appears to be described in different ways throughout the paper. It appears to me that each GP practice in the study is assigned a SMID score. This score is derived (possibly as an average) from all enrolled patients in each practice. Patients' scores are based not on their individual characteristics, but from the socio-economic profile of the areas within which they reside. If my understanding is correct, I do not think that this has been clearly articulated within the paper. Often GP clinics in deprived vs wealthy areas are referred to, when in fact, it seems to me that what is meant is GP clinics that have a high proportion of patients who live in deprived vs wealthy areas. This is important to clarify. Some GP clinics may be located in deprived areas, but have a higher number of patients living in wealthy areas. Furthermore, patients may live in wealthy areas, but experience high socioeconomic deprivation on an individual level. It is very important to clarify whether individual, area-based, or aggregate measures of socioeconomic deprivation are being used. In addition if area-level socioeconomic data on individual patients is available then this should be incorporated into any statistical models to test whether an individual's (proxy) socioeconomic position or practice socioeconomic profile has a larger impact on number of contacts and contact time. It would be better to refer to clinics as ‘practices with patients living in mostly wealthy areas’. The grouping of practices into two socioeconomic categories is also problematic and means that there is likely to be significant variation in the socioeconomic profile of practices within categories. The ‘low deprivation’ category will include practices categorised as having patients living in areas of ‘average’ deprivation (SMID 6 or 7), as will the high deprivation category (SMID 4 or 5). These practices close to the ‘average’ are likely to have different patient profiles and care experiences than practices closer to the extreme at either end (e.g SMID 1 and 2 or SMID 9 and 10). I would suggest using at least three and possibly four socioeconomic deprivation categories for analysis to avoid grouping together practices with very different socioeconomic profiles. One way could be to have Low (SMID 8, 9, 10) Medium (SMID 4, 5, 6, 7) and High (SMID 1, 2, 3) deprivation categories. I accept there will always be ‘border issues’ but having a binary category of low vs high is unacceptable if the ‘average’ levels of deprivation are included in both. It is also potentially problematic that patients who have moved practice have been excluded. This may be more likely to affect practices with a high proportion of patients living in deprived areas - as these populations are more residentially mobile. The information on who was excluded, and their socioeconomic, long-term condition, age, etc profile should be clearly reported. Rurality: There is no engagement with issues of rurality in this analysis, despite being mentioned as one of the sampling criteria. Internationally, it is recognised that rurality can exacerbate existing (socioeconomic and ethnic) health inequities and rurality should be included in any statistical models either as a control, or results should be presented stratified by rurality. It is also concerning that all appointments that were not face-to-face have been excluded from the analysis, as these are more likely to be utilised by patients living
--	--

	rurally. The rurality of each practice should be included as a factor in any statistical modelling/analysis. Ethnicity: The exclusion of information about patient ethnicity is a major limitation of this research. International evidence that access to healthcare and health outcomes are worse for ethnic minorities over and above any socioeconomic disadvantage. Therefore, at a bare minimum this should be highlighted as a very important limitation to this work, and the information about the quality and availability of patient ethnicity data should be reported (eg % missing). This will no doubt support measures to update national health data systems to accurately and consistently record ethnicity data and make it available for research purposes. Statistical analysis: Multivariate regression modelling is needed to determine which factors contribute to variation in number of contacts and contact time, and the relative importance of each factor. Also, incorporating data on individual (area-level) socioeconomic position is needed to determine whether it is practice level or individual (area) level socioeconomic deprivation that has the biggest effect on number of contacts and contact time. Individuals living in socioeconomically deprived areas who are registered with mostly wealthy practices may experience higher levels of unmet need than other patients in the same practice and vice versa. This analysis is needed to test the statement in the conclusion that “Put simply, a patient registered at a practice serving an affluent population will receive more care from their practice than a patient of the same age, sex, and level of morbidity, registered with a practice serving a more SE deprived population.” It should be stated that t-tests are for ‘statistical significance’ and not necessarily ‘importance’. It is unclear what statistical software has been used. Table 2: p level to 2 decimal places is probably fine. Both tables: using a comma between the 1000s in large numbers would improve readability In the Patient and Public Involvement section it seems that this is more representative of clinician and healthcare provider involvement. If patients and/or the public have not been involved in the design and/or undertaking of this research it should be stated clearly. I look forward to seeing a revised version of this manuscript in due course. Many thanks again for the opportunity to review this paper.
--	--

VERSION 1 – AUTHOR RESPONSE

Reviewer: 1

Comments to the Author:

lines 81 to 83, I don't understand how the conclusion reached at the end of the sentence is warranted by what goes before

We have added a few words and a reference (lines 70-71) that shows that the number of LTCs is associated with subsequent hospitalisations, whether recorded in a research setting, or derived from primary care data.

Table 1. Why not do some statistical testing here to establish whether the groups are different?

We chose not to report p-values, since we would expect the two populations to be different, and given the sample size, the differences are likely to be statistically significant. We have run chi-square tests on these numbers, and all are $p < 0.0001$. We could add these p-values, if the editor considers it necessary.

Table 2. instead of or in addition to saying "over 3 years" could the authors give the actual dates

The methods section gave the dates as September 2013 to September 2016. We have now added the actual dates at lines 91-92.

Line 204. It might be worth a sentence saying this is only measuring the unmet need of those who actually turn up at GPs, there will be other levels of unmet need in the community unable to be measured here

We have added some text to the strengths and limitation section (lines 202-206):

"However, our analysis excludes patients who made no appointments with their practice over the 3-year period. This may slightly bias our estimates of the average time spent with practice staff. However, such patients are likely to have very low health care need, and will have little impact on the resource requirements of their practices."

Reviewer: 2

Comments to the Author:

Thank you for the opportunity to review this important and interesting paper.

Overall, the paper is of a high standard and I enjoyed reading and reviewing it.

There are however some key points that need to be addressed.

Level of care: First, there are several conceptual issues that need to be clarified or appropriately addressed. One issue is that the outcomes of interest (number of contacts and total contact time) are very often conflated with "level of care" as outlined in the study objectives, or "unmet need" as described in the first sentence of the introduction. Although there is likely to be a correlation between contacts and contact time with unmet need / level and quality of care, it is important to clearly state that this paper is not directly measuring the 'level', 'quality', or 'effectiveness' of treatment provided in different practices. The big assumption of this whole paper is that "time-spent" and "number of contacts" = better quality treatment, but I'm not sure if that has been clearly articulated with sufficient supporting evidence.

Thank you for this feedback. We agree that the total time spent in contact with practice staff is not a direct measure of the quality or effectiveness of care provided, but we consider that it is a reasonable measure of the quantity of (one aspect of) health care received by each patient. Our assumption is not that more contact time equals better quality treatment, simply that those with greatest need (more long term conditions) should receive more health care. Our premise is that patients with differing socioeconomic status but with equal need (i.e. same age, same gender, and the same number of LTCs) should, on average, receive the same amount of health care as measured by contact time. One might indeed expect that patients living in less deprived areas will have a higher level of education, and health literacy, and so be better able to self-manage their conditions or comply with prescribed medications, so that if anything, practices serving less deprived populations might be able to deliver the same level of care with less contact time. However, that is beyond the scope of this paper. Here, we are simply highlighting the lack of equity that currently exists – i.e. the apparent unmet need in more socioeconomically deprived populations.

Socioeconomic position: Another key issue is that socio-economic deprivation is not clearly conceptualised, and appears to be described in different ways throughout the paper. It appears to me that each GP practice in the study is assigned a SMID score. This score is derived (possibly as an

average) from all enrolled patients in each practice. Patients' scores are based not on their individual characteristics, but from the socio-economic profile of the areas within which they reside. If my understanding is correct, I do not think that this has been clearly articulated within the paper. Often GP clinics in deprived vs wealthy areas are referred to, when in fact, it seems to me that what is meant is GP clinics that have a high proportion of patients who live in deprived vs wealthy areas. This is important to clarify. Some GP clinics may be located in deprived areas, but have a higher number of patients living in wealthy areas. Furthermore, patients may live in wealthy areas, but experience high socioeconomic deprivation on an individual level. It is very important to clarify whether individual, area-based, or aggregate measures of socioeconomic deprivation are being used. In addition if area-level socioeconomic data on individual patients is available then this should be incorporated into any statistical models to test whether an individual's (proxy) socioeconomic position or practice socioeconomic profile has a larger impact on number of contacts and contact time. It would be better to refer to clinics as 'practices with patients living in mostly wealthy areas'.

Thanks again for this feedback. It is always difficult to get the wording right. The reviewer is correct: each individual patient has a SIMD score derived from their area of residence; these are averaged across the practice populations to generate a mean SIMD score for the practice as a whole, regardless of the physical location of the practice; practices are then divided into two groups. We have edited at lines 98-103.

Our main interest was in the average health care utilisation within practices which serve more or less socioeconomically deprived populations. Broadly speaking, this is how practices are characterised when determining resource requirements in Scotland. Evaluating the health care experiences of individual patients, in relation to their socioeconomic circumstances (however measured) within the context of the practice that they are registered with, is an interesting question, but not one we were aiming to address in this paper.

We have not done any statistical modelling in the paper. Our aim is simply to highlight that GP practice systems contain information about health care need (number of long term conditions) and health care delivery (contact time) which shows a likely unmet need amongst patients served by practices with more deprived populations.

The grouping of practices into two socioeconomic categories is also problematic and means that there is likely to be significant variation in the socioeconomic profile of practices within categories. The 'low deprivation' category will include practices categorised as having patients living in areas of 'average' deprivation (SMID 6 or 7), as will the high deprivation category (SMID 4 or 5). These practices close to the 'average' are likely to have different patient profiles and care experiences than practices closer to the extreme at either end (e.g SMID 1 and 2 or SMID 9 and 10). I would suggest using at least three and possibly four socioeconomic deprivation categories for analysis to avoid grouping together practices with very different socioeconomic profiles. One way could be to have Low (SMID 8, 9, 10) Medium (SMID 4, 5, 6, 7) and High (SMID 1, 2, 3) deprivation categories. I accept there will always be 'border issues' but having a binary category of low vs high is unacceptable if the 'average' levels of deprivation are included in both.

We chose two groups to make the presentation of the data as simple as possible. Even with two groups, we were able to demonstrate the existence of an apparent unmet need in practices serving deprived populations. Our ultimate wish is that those who develop the funding allocation formulae for GP practices in Scotland (and elsewhere) should utilise these types of data, and thereby deliver a more equitable distribution of resources which reflects the health care needs of practice populations. This will require the extraction of data on a national scale, and a detailed statistical modelling exercise. If our findings are not replicated on a national scale, then so be it, but if they are, this could have a significant impact on the current funding models.

It is also potentially problematic that patients who have moved practice have been excluded. This may be more likely to affect practices with a high proportion of patients living in deprived areas - as these populations are more residentially mobile. The information on who was excluded, and their socioeconomic, long-term condition, age, etc profile should be clearly reported.

We excluded patients who left or joined each practice, or who died during the study period. These factors may well affect practices serving more deprived or affluent populations to a different extent, and these patients may also require a different level of contact with practice staff. However, for this analysis, our aim was to assess whether patients with an equal level of need for health care receive the same amount of contact time in practices serving more deprived and more affluent populations. By excluding these patients, we removed these two sources of variation in need. By only considering patients who were alive and registered with the same practice for the full 3-year period, we are better able to judge whether patients with equal need (based on age, gender, and number of LTCs) receive the same amount of primary health care.

Rurality: There is no engagement with issues of rurality in this analysis, despite being mentioned as one of the sampling criteria. Internationally, it is recognised that rurality can exacerbate existing (socioeconomic and ethnic) health inequities and rurality should be included in any statistical models either as a control, or results should be presented stratified by rurality. It is also concerning that all appointments that were not face-to-face have been excluded from the analysis, as these are more likely to be utilised by patients living rurally. The rurality of each practice should be included as a factor in any statistical modelling/analysis.

This paper considers the element of general practice access that is associated with socio-economic deprivation and this is described in the title. We agree that future research or service evaluation could investigate the overlap between socio-economic status and rurality.

Our previous analysis was focussed on face-to-face contacts so telephone contacts are excluded from the analysis. We make this explicit in the paper. The evidence for the rate of phone contacts in rural practice in Scotland pre-pandemic is patchy and therefore hard to interpret in this research context.

Ethnicity: The exclusion of information about patient ethnicity is a major limitation of this research. International evidence that access to healthcare and health outcomes are worse for ethnic minorities over and above any socioeconomic disadvantage. Therefore, at a bare minimum this should be highlighted as a very important limitation to this work, and the information about the quality and availability of patient ethnicity data should be reported (eg % missing). This will no doubt support measures to update national health data systems to accurately and consistently record ethnicity data and make it available for research purposes.

We agree and have added this to the limitations section (lines 219-220).

Statistical analysis: Multivariate regression modelling is needed to determine which factors contribute to variation in number of contacts and contact time, and the relative importance of each factor. Also, incorporating data on individual (area-level) socioeconomic position is needed to determine whether it is practice level or individual (area) level socioeconomic deprivation that has the biggest effect on number of contacts and contact time. Individuals living in socioeconomically deprived areas who are registered with mostly wealthy practices may experience higher levels of unmet need than other patients in the same practice and vice versa. This analysis is needed to test the statement in the conclusion that "Put simply, a patient registered at a practice serving an affluent population will receive more care from their practice than a patient of the same age, sex, and level of morbidity, registered with a practice serving a more SE deprived population."

Whilst we agree that multivariable modelling could be done in the way that the reviewer suggests, we deliberately chose not to go down this path. Our aim was to assess whether there was any evidence of unmet need, using data that are available in GP practice systems, but which is not utilised in determining the allocation of resources to practices within Scotland. We believe that we have achieved this through simple stratified summaries of the data. We do not think that the added complexity of regression modelling will enhance our message any further.

We feel that the statement highlighted by the reviewer is supported by the data as presented; it is not always the case that a statistical test or a regression model is necessary to make a point.

It should be stated that t-tests are for 'statistical significance' and not necessarily 'importance'.

This is true. This comment has been removed (line 143).

It is unclear what statistical software has been used.

We have added this at line 120.

Table 2: p level to 2 decimal places is probably fine.

We have changed to 2 significant figures.

Both tables: using a comma between the 1000s in large numbers would improve readability

We have made these changes.

In the Patient and Public Involvement section it seems that this is more representative of clinician and healthcare provider involvement. If patients and/or the public have not been involved in the design and/or undertaking of this research it should be stated clearly.

The RCGP Scotland P³ Committee is a lay patient group. This has been clarified in the PPI section.

I look forward to seeing a revised version of this manuscript in due course.

Many thanks again for the opportunity to review this paper.

VERSION 2 – REVIEW

REVIEWER	Graham Kirkwood Newcastle University, Institute of Health and Society
REVIEW RETURNED	13-Feb-2023

GENERAL COMMENTS	I am happy with the response to reviewers comments
--

REVIEWER	Jesse Whitehead University of Waikato
REVIEW RETURNED	05-Apr-2023

GENERAL COMMENTS	Thank you for the chance to review this paper. The point regarding the terminology "level of care" has not yet been addressed. The authors explanation in their response document makes sense, however this argument has not yet been reflected in the main text. References to the "level of care" or "quality of care/treatment" should be replaced by what the authors really mean - the "quantity of health care received by each patient" as they have explained in their response document. Thank you for clarifying how the practice socioeconomic profile was determined. I realise that my other comments were beyond the scope of your current research, but could make for interesting future studies. The limitations of only using two socioeconomic categories should be included in the limitations. This is important as much research shows a socioeconomic gradient - rather than a binary "poor" vs "rich" influence on health need/outcomes.
--

	In lieu of undertaking a statistical analysis of the differences between the excluded and the sample population, there also needs to be mention of the potential bias in the excluded population (those who moved practice during the study period). If the urban/rural spread of practices in the study is to be included as a strength of the study, then - as previously noted - more engagement with issues of rurality is required. At a minimum this would include a breakdown of the number and % of urban vs rural practices included in this study. Thank you for including ethnicity as a limitation. While it is true that regression modelling may not enhance your message further, I would argue that it is required for the concluding statement: "Put simply, a patient registered at a practice serving an affluent population will receive more care from their practice than a patient of the same age, sex, and level of morbidity, registered with a practice serving a more SE deprived population." to be considered true. However I will defer to the editors decision around this point.
--	---

VERSION 2 – AUTHOR RESPONSE

Reviewer: 1

Mr. Graham Kirkwood, Newcastle University Comments to the Author:

I am happy with the response to reviewers comments

Reviewer: 2

Dr. Jesse Whitehead, University of Waikato Comments to the Author:

Thank you for the chance to review this paper.

The point regarding the terminology "level of care" has not yet been addressed. The authors explanation in their response document makes sense, however this argument has not yet been reflected in the main text. References to the "level of care" or "quality of care/treatment" should be replaced by what the authors really mean - the "quantity of health care received by each patient" as they have explained in their response document.

Response: Fair point. Have changed "level" to "amount" in a few places (lines 22, 78, 177)

Thank you for clarifying how the practice socioeconomic profile was determined. I realise that my other comments were beyond the scope of your current research, but could make for interesting future studies.

The limitations of only using two socioeconomic categories should be included in the limitations. This is important as much research shows a socioeconomic gradient - rather than a binary "poor" vs "rich" influence on health need/outcomes.

Response: Agreed. Have added lines 221-223, and 260-263, to clarify the need for replication of our findings using more data and more detailed analyses.

In lieu of undertaking a statistical analysis of the differences between the excluded and the sample population, there also needs to be mention of the potential bias in the excluded population (those who moved practice during the study period).

Response: This was already touched on, at lines 202-206.

If the urban/rural spread of practices in the study is to be included as a strength of the study, then - as previously noted - more engagement with issues of rurality is required. At a minimum this would include a breakdown of the number and % of urban vs rural practices included in this study.

Response: We agree, and have added a couple of sentences at lines 224 and 228, to bolster this section.

Thank you for including ethnicity as a limitation.

While it is true that regression modelling may not enhance your message further, I would argue that it is required for the concluding statement: "Put simply, a patient registered at a practice serving an affluent population will receive more care from their practice than a patient of the same age, sex, and level of morbidity, registered with a practice serving a more SE deprived population." to be considered true. However I will defer to the editors decision around this point.

Response: We stratified the data by these factors, and the differences between practices serving affluent and deprived populations are starkly apparent, once LTCs are factored in, so we feel the statement is reasonable. Still, we have added a few words at line 243, to temper the language, and we do acknowledge that what we have done is simple (line 252) and needs to be replicated using more data and more detailed analysis (lines 260-263).